# GROOT: Generating Robust Watermark for Diffusion-Model-Based Audio Synthesis

Submission Id: 1178

## ABSTRACT

Amid the burgeoning development of generative models like diffusion models, the task of differentiating synthesized audio from its natural counterpart grows more daunting. Deepfake detection offers a viable solution to combat this challenge. Yet, this defensive measure unintentionally fuels the continued refinement of generative models. Watermarking emerges as a proactive and sustainable tactic, preemptively regulating the creation and dissemination of synthesized content. Thus, this paper, as a pioneer, proposes the generative **ro**bust audi**o** wa**t**ermarking method (*Groot*), presenting a paradigm for proactively supervising the synthesized audio and its source diffusion models. In this paradigm, the processes of watermark generation and audio synthesis occur simultaneously, facilitated by parameter-fixed diffusion models equipped with a dedicated encoder. The watermark embedded within the audio can subsequently be retrieved by a lightweight decoder. The experimental results highlight Groot's outstanding performance, particularly in terms of robustness, surpassing that of the leading state-of-the-art methods. Beyond its impressive resilience against individual post-processing attacks, Groot exhibits exceptional robustness when facing compound attacks, maintaining an average watermark extraction accuracy of around 95%.

## KEYWORDS

Generative audio watermarking, Proactive supervision, Text-to-Speech synthesis, Diffusion models

## 1 INTRODUCTION

The advancements in generative adversarial networks (GANs) [9, 14, 15, 47] and diffusion models (DMs) [10, 35–37] have revolutionized the way of multimedia content generation. These models have significantly reduced the gap between generated content and authentic content, blurring the lines between what is real and what is artificial. To cope with the confusion on distinguishing between natural and synthesized content, deepfake detection has explored a variety of innovative approaches to widen the distinction. Regrettably, this effort not only enhances deepfake detection techniques but also drives the evolution of generative models. Generative models (GMs) learn from these distinctions and improve further.

For several decades, watermarking has served as a proactive solution for protecting the intellectual property rights of multimedia content and tracing its origins. This technique continues to be relevant and effective in modern contexts, particularly for proactively combating deepfake threats and sourcing corresponding GMs. Recent research has identified *post-hoc watermarking* and *generative watermarking* as two primary strategies for tagging AI-generated content. Post-hoc watermarking involves embedding the watermark after the content's generation, making it an asynchronous

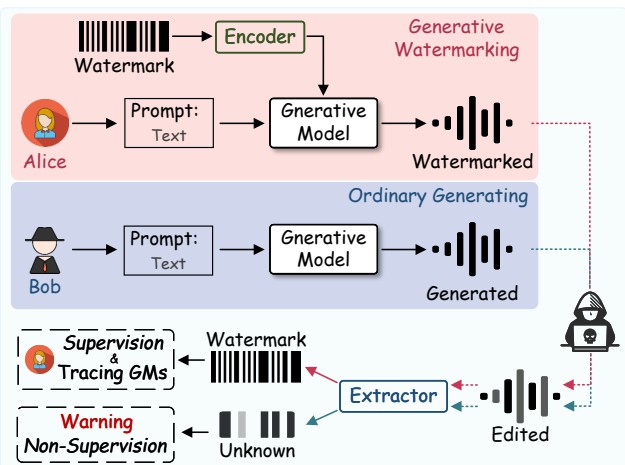

**Figure 1: The diagram illustrates the process of supervising generated content through generative watermarking. The synthetic content generated via GMs by Alice will be subject to regulation, while Bob's may pose a high risk to society.**

process. In contrast, generative watermarking integrates the watermarking process with content synthesis, utilizing the same GM for both tasks. Specifically, in terms of generative image watermarking, one of the solutions incorporates watermarks into the weights or structure of GMs [22, 43, 46]. Another explores the transferability of watermarks [8, 23, 28, 40, 42, 45], where the watermark is applied to the training data and consequently embedded in all generated images, leveraging its transferable characteristics.

Generative image watermarking [8, 42, 43] has rapidly gained prominence among researchers due to its superior fidelity and robustness. While this field flourishes within the realm of imagery, generative audio watermarking remains underexplored and lacks similar advancements. Chen et al. [3] and Ding et al. [7] pioneered the exploration of generative steganography in speech using autoregressive models. Building on this foundation, Cho et al. [5] and Li et al. [21] introduced generative audio steganography algorithms based on GANs. Designed to address particular applications such as model attribution and coverless steganography, these algorithms operate under the assumption that the distribution of generated content occurs through lossless channels-a condition that deviates significantly from real-world scenarios. The inherent limitation in robustness constrains the ability of these algorithms to proactively regulate the utilization of generated content. Furthermore, advancements of DMs have accelerated the audio synthesis, leading to their widespread application. However, the aforementioned research has not yet explored the potential of generative watermarking based on DMs.

To tackle these challenges, we proposed a generative **ro**bust audi**o** wa**t**ermarking (**Groot**) method tailored for diffusion-model-based audio synthesis. By directly generating watermarked audio

through DMs, we can regulate and trace the use conditions of the generated audio and its originating DMs. As illustrated in Fig. 1, Alice's route demonstrates the conciseness but effectiveness of our proposed Groot method in comparison to the ordinary audio generation process, depicted as Bob's route. Specifically, the encoder converts the watermark into a format recognizable by DMs. Following joint optimization with a carefully designed loss function, the DMs can directly produce watermarked audio from the input watermark. A precise decoder is then employed to accurately extract the watermark from the generated audio. Our approach marries generative watermarking with proactive supervision, with the training overhead being exclusive to the encoder and decoder. This eliminates the necessity for complex retraining of DMs. Such a feature makes our method versatile and readily implementable as a plug-and-play solution for any diffusion model.

In a nutshell, the contributions can be summarized as:

- **New Paradigm.** We pioneered an exploratory investigation and proposed a generative audio watermarking technique for proactively supervising generated content and tracing its originating models. We utilized a meticulously designed watermark encoder and decoder to directly synthesize watermarked audio through the diffusion models.
- **Vigorous Robustness.** The robustness experiments validate that Groot exhibits remarkable resilience against not only individual post-processing attacks but compound attacks formed by arbitrary combinations of single attacks.
- **High Performance.** We empirically validate several criteria, encompassing fidelity and capacity. It illustrates Groot maintains a superior quality in watermarked audio and can adapt to large capacities of up to 5000 bps.

## 2 RELATED WORK

### 2.1 Text-to-Speech Diffusion Models

Text-to-speech (TTS) is a technique that synthesizes the waveform from the transcriptions. Nowadays, TTS embraces its significant performance boost by relying on the advantage of diffusion models. A complete TTS synthesis process consists of two main stages designed as deep neural networks: *text-to-spectrogram* [2, 12, 27, 31] and *spectrogram-to-waveform* [4, 11, 18–20, 26]. The methods employed for text-to-spectrogram leverage diffusion models to generate the mel-spectrogram from Gaussian noise, with text input serving as the prompt. For spectrogram-to-waveform, these approaches synthesize the waveform by utilizing the mel-spectrogram as conditional input to the DMs. The proposed Groot primarily leverages the spectrogram-to-waveform process for watermarking. DMs generate watermarked audio directly by taking the watermark transformed into the latent variable as input.

### 2.2 Audio Watermarking

**Deep-Learning-based Watermarking** With the gradual advancement of deep learning in the field of audio watermarking, an increasing number of deep-learning-based audio watermarking techniques have emerged [1, 24, 25, 30, 33]. Concretely, Chen et al. [1] utilized invertible neural networks (INNs) for embedding the watermark into audio to boost robustness. Liu et al. [24] pioneered a watermarking framework to withstand audio re-recording. To

detect speech generated by AI, Roman et al. [33] devised a localized watermarking technique. In addition, Liu et al. [25] proposed timbre watermarking, aiming to defeat the voice cloning attacks.

**Generative Watermarking** Diverging from post-hoc watermarking techniques, generative audio watermarking (or steganography) [3, 5, 7, 21] roots the watermark (or secret message) into GMs, facilitating the direct synthesis of watermarked (or stego) audio. Both [3] and [7] leverage autoregressive models to generate realistic cover speech samples. Specifically, [3] utilizes adaptive arithmetic decoding (AAD), whereas [7] employs the distribution copy method for embedding the secret message. Utilizing GANs, [21] directly generated stego audio from secret audio. For model attribution, [5] also employs GANs, which are trained to synthesize watermarked speech by incorporating a specific key and constraints. While these methods are constrained by transmission channels and limitations in model and practical application, the proposed Groot stands out as a plug-and-play generative watermarking method, enabling supervision through extracted watermarks. Moreover, it is capable of adapting to more robust scenarios.

## 3 PRELIMINARIES

The proposed Groot leverages vocoders based on DDPM [10] to generate the watermarked audio. The blurb of diffusion-model-based vocoders about generation is described as follows.

In the forward diffusion process, the normally distributed input $\mathbf{s}_T$ is produced by gradually adding Gaussian noise to the original audio $\mathbf{s}_0 \sim q_{data}(\mathbf{s}_0)$. It follows a discrete Markov chain $\{\mathbf{s}_t\}_{t=0}^T$, which is also Gaussian distributed, that is

$$q(\mathbf{s}_t|\mathbf{s}_{t-1}) = \mathcal{N}(\mathbf{s}_t; \sqrt{1-\beta_t}\mathbf{s}_{t-1}, \beta_t\mathbf{I}), \quad (1)$$

where $\beta_t \in (0, 1)$ is the variance scheduled at time step $t$, and $\mathbf{I}$ is an identity matrix. Let $\alpha_t = 1 - \beta_t$, $\overline{\alpha}_t = \prod_{i=1}^t \alpha_i$ and $\epsilon \sim \mathcal{N}(\mathbf{0}, \mathbf{I})$. For any time step of $t$, by re-parameterization, one has $\mathbf{s}_t = \sqrt{\overline{\alpha}_t}\mathbf{s}_0 + \sqrt{1-\overline{\alpha}_t}\epsilon$.

The reverse denoising process generates an estimate of the audio waveform from the input $\mathbf{s}_T$ through a UNet-like network $\epsilon_\theta$. Given that the denoising process $q(\mathbf{s}_{t-1}|\mathbf{s}_t)$ follows a Gaussian distribution, according to Bayes' theorem, we can derive $q(\mathbf{s}_{t-1}|\mathbf{s}_t)$ as follows:

$$q(\mathbf{s}_{t-1}|\mathbf{s}_t) = \frac{q(\mathbf{s}_t|\mathbf{s}_{t-1})q(\mathbf{s}_{t-1}|\mathbf{s}_0)}{q(\mathbf{s}_t|\mathbf{s}_0)}. \quad (2)$$

Unfold this equation with Eq. 1 and combine like terms, we get

$$q(\mathbf{s}_{t-1}|\mathbf{s}_t) = \mathcal{N}\big(\mathbf{s}_{t-1}; \frac{1}{\sqrt{\alpha_t}}(\mathbf{s}_t - \frac{1-\alpha_t}{\sqrt{1-\overline{\alpha}_t}}\epsilon), (\frac{1-\overline{\alpha}_{t-1}}{1-\overline{\alpha}_t}\beta_t)\mathbf{I}\big). \quad (3)$$

The network $\epsilon_\theta$ is trained to estimate the noise $\epsilon$. Once it has been trained well, the estimated audio can be obtained by $\epsilon_\theta$ with re-parameterization:

$$\mathbf{s}_{t-1} = \frac{1}{\sqrt{\alpha_t}}\left(\mathbf{s}_t - \frac{1-\alpha_t}{\sqrt{1-\overline{\alpha}_t}}\epsilon_\theta(\mathbf{s}_t, t, c)\right) + \delta_t\mathbf{z}, \quad (4)$$

where $\delta_t^2 = \frac{1-\overline{\alpha}_{t-1}}{1-\overline{\alpha}_t}\beta_t\mathbf{I}$, $\mathbf{z} \sim \mathcal{N}(\mathbf{0}, \mathbf{I})$, and $c$ is the mel-spectrogram.

## 4 METHODOLOGY

Our proposed method, Groot, aims to seamlessly connect the input latent variables of DMs with the generation of watermarked audio content. To achieve this, we are confronted with three primary challenges: first, designing an architecture to incorporate the

Figure 2: The pipeline of Groot. a) Training Process, where the watermark w is encoded into a latent variable $\sigma$ using the encoder E($\cdot$). A Gaussian latent variable $s_T$ is sampled from a standard distribution. Watermarked audio is then generated from the final latent variable by adding $s_T$ and $\sigma$ via diffusion models, employing the mel-spectrogram as a condition. The extracting stage employs the watermark decoder D($\cdot$) to recover the watermark $\hat{w}$ from the watermarked audio. b) Inference Process, where the watermark from the encoder is directly used by the diffusion model to synthesize the watermarked audio, eliminating the need for additional Gaussian latent variables.

watermark into the input of DMs; second, developing a training approach that encourages DMs to naturally generate watermarked content; and third, devising a reliable architecture for extracting the watermark from the generated audio.

To solve the question above, solutions encompassing three main phases-watermarking, generating, and extracting-are illustrated in Fig. 2. Firstly, an encoder E($\cdot$) is contrived to transform the watermark w into a latent variable. This transformed watermark is then combined with the origin latent variable $s_T$, serving as the input of DMs. The diffusion model employed is a pre-trained architecture, fixed and solely dedicated to generating audio content embedded with the watermark. Subsequently, an advanced decoder D($\cdot$) is developed to accurately extract the watermark from the synthetic audio. To ensure DMs incorporate the watermark into the audio content, we define a loss function that facilitates a joint training process. Consequently, the encoder E($\cdot$), a diffusion model(with its parameters fixed), and the decoder D($\cdot$) are simultaneously trained, guided by the criteria set forth by the loss function.

Building upon the three main phases previously outlined, the subsequent section will detail the specific architectures of both the watermark encoder and decoder. Additionally, the methodologies for embedding and extracting watermarks, the approach to joint training, and the theory for watermark verification will be meticulously described, each in their respective segments.

## 4.1 Watermark Encoder and Decoder

The watermark encoder aims to convert the watermark w to a latent variable $\sigma$, which satisfies the distribution of DMs' input.

The purpose of the watermark decoder is to distinguish features between the audio and the watermark, extracting the watermark $\hat{w}$ from the watermarked audio $x_0$.

The *watermark encoder* is mainly composed of linear layers and Rectified Linear Unit (ReLU) activation functions. As depicted in Fig. 3, outputs of Fully Connected (FC) layers need to thoroughly get through the ReLU activation function. The input size of the first FC layer, namely the length of w is configurable, while the output of the last FC layer needs to adapt for the input size of DMs. Under the specific layout of the encoder, the watermark is converted to a latent variable $\sigma$ with Gaussian distribution, which can be recognized as an input for diffusion models.

The *watermark decoder* is designed with a combination of a convolutional block (ConvBlock) and a Dense Block as illustrated in Fig. 3. The ConvBlock consists of seven *modified gated convolutional neural networks* (MGCNN), each intricately designed to enhance the model's feature extraction capabilities. The architecture then transitions into a Dense Block, which is meticulously structured beginning with a FC layer, followed by a ReLU activation for nonlinear transformation. This sequence is succeeded by another FC layer, setting the stage for the final component of the model. The culmination of the process is marked by the application of a Sigmoid activation function, strategically chosen to predict the output with precision. This comprehensive configuration ensures a robust and efficient process, optimized for high-performance on robustness.

The designed MGCNN represents an advanced iteration of the gated convolution neural networks (GCNN) [6]. This model is structured into two distinct branches: The first process data through

**Figure 3: Architecture of the Encoder and Decoder.**

a singular convolutional layer, ensuring direct feature extraction. In parallel, the second branch initially passes through a convolutional layer to extract features, which is immediately followed by batch normalization to ensure data standardization and improve network stability. The sequence culminates with the application of a sigmoid activation function, a strategic choice that enables the implementation of an effective gating mechanism. The gating mechanism stands as a cornerstone within the model, functioning to meticulously regulate the flow of vital information to subsequent layers. This strategic regulation is pivotal in effectively addressing and mitigating the vanishing gradient problem.

## 4.2 Watermark Embedding and Extracting

*Watermark embedding:* During the embedding process, the Gaussian latent variable $\mathbf{s}_T$, sampled from the standard Gaussian distribution, continues to serve as the input for DMs. To root the watermark into DMs, the watermark encoder $\mathbf{E}(\cdot)$ is utilized to transform the watermark $\mathbf{w}$ into the latent variable $\sigma$ before generation process:

$$\sigma = \mathbf{E}(\mathbf{w}) \in \mathbb{R}^v, \tag{5}$$

where $v = b \times l_s$, $b$ is the batch size and $l_s$ represents the length of the Gaussian latent variable $\mathbf{s}_T$. Subsequently, the latent variable $\sigma$ is superposed to $\mathbf{s}_T$ to acquire the final latent variable $\mathbf{x}_T$ for generating watermarked audio:

$$\mathbf{x}_T = \mathbf{s}_T + \sigma. \tag{6}$$

The generation process for watermarked audio $\mathbf{x}_0$ is completed through the denoising process, directly derived from the final latent variable $\mathbf{x}_T$. The denoising distribution of this generation process adheres to the same format of Eq. (3), with the input being replaced by $\mathbf{x}_T$. In a consequence, the estimated audio can be obtained by

$$\mathbf{x}_{t-1} = \frac{1}{\sqrt{\alpha_t}}\left(\mathbf{x}_t - \frac{1 - \alpha_t}{\sqrt{1 - \overline{\alpha}_t}}\epsilon_\theta(\mathbf{x}_t, t, c)\right) + \delta_t \mathbf{z}, \tag{7}$$

where the pretrained noise prediction network $\epsilon_\theta(\mathbf{x}_t, t, c)$ is utilized to approximate the denoising distribution. Here, $c$ represents the mel-spectrogram, acting as a conditional input to assist in generating watermarked audio. And $\delta_t \mathbf{z}$ represents the random noise, introduced to enhance randomness in the generation process and augment the diversity of the audio. Finally, a traditional sampler is employed to generate watermarked audio step by step. The intact watermarking and generating stage are formalized in Algorithm 1.

*Watermark extraction:* The watermark extraction process is an independent process that does not necessitate the use of diffusion and denoising process. The watermark decoder $\mathbf{D}(\cdot)$ is designed to disentangle features between audios and watermarks for recovering

the watermark $\hat{\mathbf{w}}$:

$$\hat{\mathbf{w}} = \mathbf{D}(\mathbf{x}_0) \in \mathbb{R}^u, \tag{8}$$

where $u = b \times l$, $b$ denotes the batch size, $l$ denotes the length of the watermark. Our approach enables precise supervision of generated content and corresponding DMs by ensuring that the extracted watermark $\hat{\mathbf{w}}$ aligns with the embedded watermark. Essentially, through this innovative watermarking and extracting progress, the watermark becomes seamlessly integrated into the audio, maintaining its perceptual quality while ensuring content authenticity and traceability.

During the *inference process*, the process of embedding watermarks is streamlined compared to the training process. Unlike the training procedure, where the latent variable $\sigma$, derived from the watermark encoder, is added to the Gaussian latent variable $\mathbf{s}_T$, for inference, $\sigma$ is directly fed into the diffusion model. Thus, as a coverless watermarking technique, our method allows for direct use of the watermark $\mathbf{w}$ as input to the pretrained DMs, as illustrated in Fig. 2. This can be formulated as:

$$\mathbf{x}_0 = \mathcal{G}(\mathbf{E}(\mathbf{w})), \tag{9}$$

where $\mathcal{G}(\cdot)$ symbolizes the generative progress employed by DMs. Ultimately, the pretrained watermark decoder skillfully extracts the watermark $\hat{\mathbf{w}}$ from the watermarked audio.

---

**Algorithm 1:** Watermarking and Generating Stage.

**Input:** Watermark $\mathbf{w}$, Mel-spectrogram $c$.
**Output:** Watermarked audio $\mathbf{x}_0$

1   $\mathbf{s}_T \leftarrow \mathbf{s}_T \sim \mathcal{N}(0, \mathbf{I})$;
2   $\sigma \leftarrow \mathbf{E}(\mathbf{w})$;          $\triangleright \mathbf{w} = \{(w_i), w_i \in \{0,1\}\}_{i=1}^l$
3   $\mathbf{x}_T \leftarrow \mathbf{s}_T + \sigma$;               $\triangleright$ Watermarking
4   **for** $t \leftarrow T_{infer}, ..., 1$ **do**
5      **if** $t < 2$ **then** $\mathbf{z} \leftarrow 0$ **else** $\mathbf{z} \sim \mathcal{N}(0, \mathbf{I})$;
6      $\mathbf{x}_{t-1} \leftarrow \frac{1}{\sqrt{\alpha_t}}\left(\mathbf{x}_t - \frac{1-\alpha_t}{\sqrt{1-\overline{\alpha}_t}}\epsilon_\theta(\mathbf{x}_t, t, c)\right) + \delta_t \mathbf{z}$;
7   **end**
8   **return** $\mathbf{x}_0$

---

## 4.3 Jointly Optimization of Training Process

Our Groot method places a paramount emphasis on the dual objectives of maintaining high-quality watermarked audio and achieving precise extraction of watermarks. To adeptly balance these critical aspects, we have implemented a joint optimization strategy. This approach focuses on conducting gradient updates for both the watermark encoder and decoder, while strategically keeping the parameters of DMs unchanged. Regarding the quality of watermarked audio, we employ the logarithm Short-Time Fourier Transform (STFT) magnitude loss $\mathcal{L}_{mag}(\mathbf{s}_0, \mathbf{x}_0)$ as in [41]. It utilizes an $L_1$ norm to constrain the log-magnitude, as defined by:

$$\mathcal{L}_{mag} = ||\log(\mathbf{STFT}(\mathbf{s}_0)) - \log(\mathbf{STFT}(\mathbf{x}_0))||_1, \tag{10}$$

where $||\cdot||_1$ denotes the $L_1$ norm, $\mathbf{STFT}(\cdot)$ represents the STFT magnitudes, $\mathbf{s}_0$ and $\mathbf{x}_0$ represent the original audio and watermarked audio respectively. Furthermore, drawing upon [17], we integrate the mel-spectrogram loss $\mathcal{L}_{mel}(\mathbf{s}_0, \mathbf{x}_0)$, which measures the distance between the original and watermarked audio using the $L_1$ norm. It is expressed as:

$$\mathcal{L}_{mel} = \mathbf{E}_{(\mathbf{s}_0, \mathbf{x}_0)}\left[||\phi(\mathbf{s}_0) - \phi(\mathbf{x}_0)||_1\right], \tag{11}$$

where $\phi(\cdot)$ denotes the function of mel-spectrogram transformation. The total loss that constrains watermarked audio quality can be computed as:

$$\mathcal{L}_{Aud} = \lambda_{mag}\mathcal{L}_{mag} + \lambda_{mel}\mathcal{L}_{mel}, \tag{12}$$

where $\lambda_{mag}$ and $\lambda_{mel}$ are hyper-parameters for the log STFT magnitude loss and mel-spectrogram loss, respectively. Their target is to strike a balance between these loss terms.

To guarantee successful watermark recovery, binary cross-entropy stand as the indispensable choice:

$$\mathcal{L}_{WM} = -\sum_{i=1}^{k} w_i \log \hat{w}_i + (1 - w_i) \log(1 - \hat{w}_i). \tag{13}$$

Our final training loss is structured as follows, with both the watermark encoder and decoder optimized synchronously,

$$\mathcal{L} = \tau\mathcal{L}_{Aud} + \mathcal{L}_{WM}, \tag{14}$$

where $\tau$ is the hyper-parameters for the total audio quality loss $\mathcal{L}_{Audio}$, used to control the trade-off between audio quality and watermark extraction accuracy.

### 4.4 Watermark Verification

Regulating generated contents and tracing associated DMs are achieved by verifying the existence of the watermark within the generated audio through test hypothesis [23, 43]. By assuming the watermark bit errors are independent to each other, with the previously defined watermark bits length $l$, the number of matching watermark bits $\kappa$ follow the binomial distribution:

$$Pr(X = \kappa) = \sum_{i=\kappa}^{l} \binom{l}{i} \xi^i (1 - \xi)^{l-i}, \tag{15}$$

where $\xi$ is the probability that needs to be tested under hypotheses. In the common use of the binomial test, the null hypothesis $H_0$ states the variable $X$ is observed with a random guess of probability $\xi = 0.5$, whereby the model is not watermarked. The alternative hypothesis $H_1$ states the watermark is produced by the owner.

We determine a threshold $T_S$ with a given false positive rate (FPR) which is calculated from the probability density function under $H_0$. Alternatively, if $\xi$ is sufficiently large under $H_1$, the false negative rate (FNR) will be low, and FNR can be used to assess whether the chosen threshold provides a satisfactory trade-off in distinguishing between false positive cases and missed detection cases.

To exemplify the hypothesis test, a successful verification in our experiments is given. In our case, the total number of samples is 768 and the length of the watermark is $l$=100. Then, we set an accepted FPR≤0.0037. From the experimental result, we have $\xi$=0.5442 and $\xi$=0.9969 under $H_0$ and $H_1$, and the corresponding distributions are plotted in Fig. 4. From the false cases in the figure, we find the threshold $T_S \in [0.61, 0.99]$. With the threshold, FNR≤0.012. Such a small FNR indicates that the detected cases with $\xi$=0.9963 are worth trusted. In simpler terms, when the watermark is 100 bits long, the extraction accuracy of 99.63% can be utilized to confirm the existence of the watermark.

## 5 EXPERIMENTAL RESULTS AND ANALYSIS

In this section, we conduct comprehensive experiments to rigorously evaluate our Groot method across several dimensions: fidelity and capacity, robustness. Moreover, we undertake a comparative

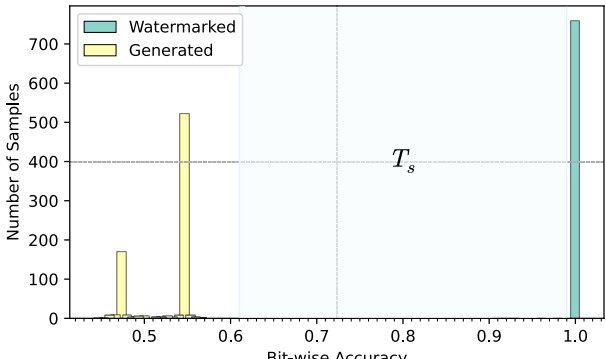

**Figure 4: A test result for binomial distribution under hypothesis $H_0$ and $H_1$ for $\xi$ =0.5442 and $\xi$ = 0.9969. Given the total number of samples is 768, an FPR≤0.0037 and the threshold will be $T_S \in [0.61, 0.99]$. With the threshold, FNR≤0.012.**

analysis of Groot against the current state-of-the-art (SOTA) methods. The details of these experiments and their respective analyses are detailed below.

### 5.1 Experimental Setup

*5.1.1 Dataset and Baseline.* Our experiments were carried out on LJSpeech [13], LibriTTS [44], and LibriSpeech [29] datasets. LJSpeech is a single-speaker English speech dataset featuring approximately 24 hours of audio data with a sample rate of 22.05 kHz. LibriTTS and LibriSpeech both are multi-speaker English datasets comprising around 584 hours of audio data recorded at a sample rate of 24kHz and approximately 1000 hours of audio recordings with a sample rate of 16kHz, respectively. In addition, we conducted a comprehensive comparative evaluation of proposed Groot against existing SOTA methods, including WavMark [1], DeAR [24], AudioSeal [33], and TimbreWM [25].

*5.1.2 Evaluation Metrics.* We evaluated the performance of our method with different objective evaluation metrics. Short-Time Objective Intelligibility (STOI) [38] predicts the intelligibility of audio. Mean Opinion Score of Listening Quality Objective (MOSL) assesses audio quality based on the Perceptual Evaluation of Speech Quality [32]. We also conducted evaluation metrics using Structural Similarity Index Measure (SSIM) [39]. Moreover, Bit-wise Accuracy (ACC) is employed to evaluate watermark extraction accuracy.

*5.1.3 Implementation Details.* 1) *Model settings.* We conducted validation on Groot utilizing DiffWave [18], WaveGrad [4], and PriorGrad [20]. DiffWave [18] and WaveGrad [4] both are vocoders that utilize accelerated sampling and employ a gradient-based sampler akin to Langevin dynamics for audio synthesis, respectively. To boost the computational efficiency of vocoders, PriorGrad [20] leverages an adaptive prior derived from data statistics, which are conditioned on the provided conditional information. 2) *Training settings.* During the training process, the Adam optimizer [16] was utilized to update the parameters, and the learning rate was set to 2e-4. Regarding the hyper-parameters of audio quality loss in Eq. (12), we empirical set $\lambda_{mag} = 0.7$ and $\lambda_{mel} = 0.3$. All experiments were performed on the platform with Intel(R) Xeon Gold 5218R CPU and NVIDIA GeForce RTX 3090 GPU.

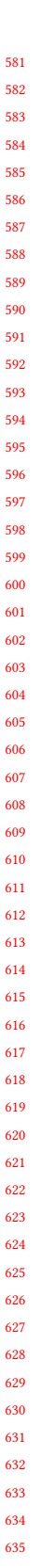

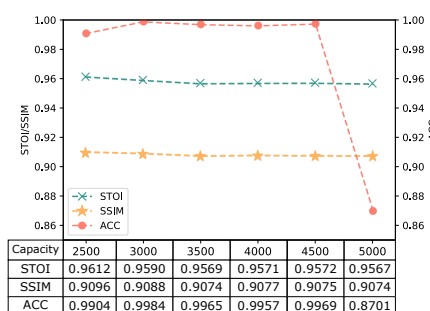

| Capacity | 2500 | 3000 | 3500 | 4000 | 4500 | 5000 |
|---|---|---|---|---|---|---|
| STOI | 0.9612 | 0.9590 | 0.9569 | 0.9571 | 0.9572 | 0.9567 |
| SSIM | 0.9096 | 0.9088 | 0.9074 | 0.9077 | 0.9075 | 0.9074 |
| ACC | 0.9904 | 0.9984 | 0.9965 | 0.9957 | 0.9969 | 0.8701 |

(a) LJSpeech

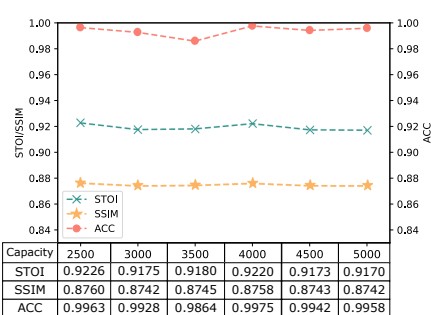

| Capacity | 2500 | 3000 | 3500 | 4000 | 4500 | 5000 |
|---|---|---|---|---|---|---|
| STOI | 0.9226 | 0.9175 | 0.9180 | 0.9220 | 0.9173 | 0.9170 |
| SSIM | 0.8760 | 0.8742 | 0.8745 | 0.8758 | 0.8743 | 0.8742 |
| ACC | 0.9963 | 0.9928 | 0.9864 | 0.9975 | 0.9942 | 0.9958 |

(b) LibriTTS

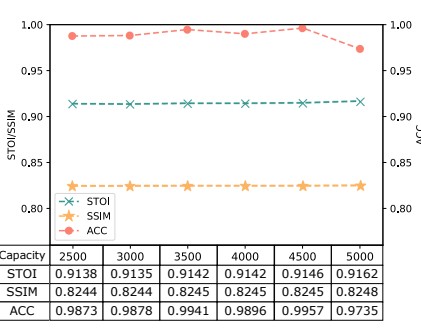

| Capacity | 2500 | 3000 | 3500 | 4000 | 4500 | 5000 |
|---|---|---|---|---|---|---|
| STOI | 0.9138 | 0.9135 | 0.9142 | 0.9142 | 0.9146 | 0.9162 |
| SSIM | 0.8244 | 0.8244 | 0.8245 | 0.8245 | 0.8245 | 0.8248 |
| ACC | 0.9873 | 0.9878 | 0.9941 | 0.9896 | 0.9957 | 0.9735 |

(c) LibriSpeech

Figure 5: The Analysis of Capacity Under Various Datasets.

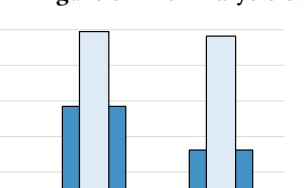

Figure 6: Ablation Study. Comparison of accuracy employing different architecture of watermark decoder.

Table 1: Fidelity of Groot. Benchmark represents generated audio. ↑ indicates a higher value is more desirable.

| Dataset | | Benchmark | Capacity (*bps*) | | | |
|---|---|---|---|---|---|---|
| | | | 100 | 500 | 1000 | 2000 |
| LJSpeech | STOI↑ | 0.9655 | 0.9605 | 0.9624 | 0.9568 | 0.9578 |
| | MOSL↑ | 3.5120 | 3.3871 | 3.4300 | 3.3432 | 3.3663 |
| | SSIM↑ | 0.8453 | 0.9088 | 0.9100 | 0.9075 | 0.9082 |
| | ACC↑ | N/A | 0.9969 | 0.9910 | 0.9979 | 0.9929 |
| LibriTTS | STOI↑ | 0.9337 | 0.9175 | 0.9166 | 0.9175 | 0.9179 |
| | MOSL↑ | 2.8159 | 2.7267 | 2.7241 | 2.7335 | 2.7353 |
| | SSIM↑ | 0.8025 | 0.8742 | 0.8740 | 0.8743 | 0.8743 |
| | ACC↑ | N/A | 0.9957 | 0.9958 | 0.9979 | 0.9961 |
| LibriSpeech | STOI↑ | 0.9176 | 0.9276 | 0.9145 | 0.9140 | 0.9136 |
| | MOSL↑ | 2.7788 | 2.7775 | 2.9271 | 2.7799 | 2.7722 |
| | SSIM↑ | 0.6699 | 0.8243 | 0.8280 | 0.8244 | 0.8244 |
| | ACC↑ | N/A | 0.9945 | 0.9898 | 0.9952 | 0.9832 |

## 5.2 Ablation Study

To assess the efficacy of MGCNN as the foundational architecture of the watermark decoder, as detailed in Section 4.1, we conducted an ablation study with LJSpeech dataset by substituting all MGCNNs with conventional CNNs as a baseline. This investigation explored a range of capacities, specifically 100, 500, 1000, 2000 bits per second (bps). Fig. 6 showcases the watermark recovery accuracy employing different decoder structures, presented through a nested bar chart. The results indicate that MGCNNs outperform traditional CNNs significantly, showcasing a substantial enhancement in recovery accuracy as depicted in the bar chart. Across various capacities, the average accuracy attained with MGCNNs is an impressive 99.47%. Conversely, employing standard CNNs in the watermark decoder leads to a marked reduction to an average of just 89.31%. Specifically, at the 100 bps capacity, the accuracy of CNNs' is 11.14% lower compared to the performance with MGCNNs.

Table 2: Fidelity of Groot under various diffusion models. Benchmark represents generated audio. ↑ indicates a higher value is more desirable.

| Diffusion Model | | Benchmark | Capacity (*bps*) | | | |
|---|---|---|---|---|---|---|
| | | | 100 | 500 | 1000 | 2000 |
| WaveGrad [4] | STOI↑ | 0.9363 | 0.8932 | 0.8947 | 0.8936 | 0.8938 |
| | MOSL↑ | 2.2339 | 2.1631 | 2.1355 | 2.1586 | 2.1251 |
| | SSIM↑ | 0.7448 | 0.7368 | 0.7300 | 0.7329 | 0.7288 |
| | ACC↑ | N/A | 0.9981 | 0.9971 | 0.9956 | 0.9925 |
| PriorGrad [20] | STOI↑ | 0.9722 | 0.9580 | 0.9579 | 0.9582 | 0.9573 |
| | MOSL↑ | 3.8875 | 2.6207 | 2.4198 | 2.5737 | 2.4665 |
| | SSIM↑ | 0.9032 | 0.8528 | 0.8328 | 0.8478 | 0.8392 |
| | ACC↑ | N/A | 0.9996 | 0.9987 | 0.9973 | 0.9962 |

Table 3: Comparison of fidelity. "D-" presents the difference values. ↑/↓ indicates a higher/lower value is more desirable.

| Capacity | Method | D-STOI↓ | D-MOSL↓ | D-SSIM↓ | ACC↑ |
|---|---|---|---|---|---|
| 16 bps | AudioSeal [33] | **0.0015** | 0.0546 | 0.0189 | 0.9214 |
| | **Groot(DiffWave)** | 0.0042 | 0.0928 | **-0.0643** | **0.9945** |
| 32 bps | WavMark [1] | **0.0003** | 0.1811 | 0.0310 | **1.0000** |
| | **Groot(DiffWave)** | 0.0023 | 0.0632 | **-0.0649** | 0.9974 |
| 100 bps | DeAR [24] | 0.2563 | 0.3904 | 0.2780 | **1.0000** |
| | TimbreWM [25] | 0.0147 | 0.6086 | 0.0612 | 0.9998 |
| | **Groot(DiffWave)** | **0.0005** | 0.1249 | **-0.0635** | 0.9969 |
| 2500 bps | DeAR [24] | 0.1497 | 1.4560 | 0.1999 | 0.5007 |
| | TimbreWM [25] | 0.2415 | 3.1842 | 0.3254 | 0.4995 |
| | **Groot(DiffWave)** | **0.0043** | **0.0861** | **-0.0643** | 0.9904 |
| | **Groot(WaveGrad)** | 0.0438 | 0.0785 | **-0.0765** | 0.9970 |
| | **Groot(PriorGrad)** | 0.0142 | 1.3182 | 0.0556 | **0.9981** |

## 5.3 Fidelity and Capacity

*5.3.1 The Performance of Proposed Groot on Fidelity and Capacity.* **Fidelity** gauges the extent to which watermarking minimally impacts the perceptibility of the original generation. We validated the fidelity of watermarked audio with the evaluation metrics exhibited in Section 5.1.2. *Table 1 elaborate the experimental results on fidelity across different datasets employing DiffWave.* In this context, *Benchmark* refers to the generated audio, whose results are compared to the *Ground Truth*. In scenarios involving *watermarked audio*, the results are presented in comparison to the *Benchmark*. The experimental results demonstrate that the quality of watermarked audio remains impressively high across various datasets For LJSpeech, the STOI metric consistently holds at 0.96, and the lowest MOSL surpasses 3.3326 with only minimal reduction. While a slight numerical decline is observed in the multi-speaker LibriTTS and LibriSpeech datasets, attributed to resampling, this decrease does not materially affect the overall quality of the watermarked audio. Importantly, there is no discernible downward trend in audio quality as the capacity increases, even at a capacity of 2000

**Table 4: Comparison of Robustness Under Individual Attacks.**

| Method | | Noise | | | LP-F | BP-F | Stretch | Cropping | | Echo |
|---|---|---|---|---|---|---|---|---|---|---|
| | | 5 dB | 10 dB | 20 dB | 3k | 0.3-8k | 2 | front | behind | default |
| WavMark [1] | STOI↑ | 0.8128 | 0.9997 | 0.9733 | 0.9996 | 0.9149 | 0.9987 | 0.4185 | 0.5135 | 0.6122 |
| | MOSL↑ | 1.1016 | 4.4625 | 2.1236 | 2.8870 | 1.1815 | 1.8689 | 1.7090 | 1.7466 | 1.3716 |
| | ACC↑ | 0.5121 | 0.5295 | 0.6523 | **0.9999** | 0.9995 | 0.9926 | 0.9797 | 0.9713 | 0.8668 |
| DeAR [24] | STOI↑ | 0.7528 | 0.7577 | 0.7551 | 0.7425 | 0.7516 | 0.7361 | 0.7511 | 0.7417 | 0.7580 |
| | MOSL↑ | 4.3118 | 4.3069 | 4.3015 | 4.2523 | 4.2923 | 4.2133 | 4.2766 | 4.1764 | 4.2655 |
| | ACC↑ | 0.7546 | 0.9076 | **1.0000** | 0.5203 | **1.0000** | **1.0000** | 0.7167 | 0.7184 | 0.9654 |
| AudioSeal [33] | STOI↑ | 0.8411 | 0.9110 | 0.9789 | 0.9978 | 0.8575 | 0.9986 | 0.4150 | 0.5348 | 0.7563 |
| | MOSL↑ | 1.0415 | 1.0995 | 1.5987 | 4.4728 | 3.7994 | 3.1729 | 1.0916 | 1.1661 | 1.1845 |
| | ACC↑ | 0.6048 | 0.6086 | 0.6600 | 0.7498 | 0.9164 | 0.8958 | 0.7226 | 0.8925 | 0.7277 |
| TimbreWM [25] | STOI↑ | 0.8797 | 0.9136 | 0.9812 | 0.9997 | 0.8579 | 0.9999 | 0.4135 | 0.5331 | 0.7559 |
| | MOSL↑ | 1.1871 | 1.3347 | 2.7424 | 4.5496 | 1.5119 | 2.5169 | 1.8149 | 1.8155 | 1.4720 |
| | ACC↑ | 0.5627 | 0.6335 | 0.8154 | 0.9934 | 0.9883 | 0.9448 | **0.9888** | **0.9814** | 0.9471 |
| **Groot(Diffwave)** | STOI↑ | 0.7789 | 0.8754 | 0.9672 | 0.9984 | 0.8515 | 0.9986 | 0.9420 | 0.9753 | 0.9166 |
| | MOSL↑ | 1.0353 | 1.0773 | 1.4905 | 4.6186 | 3.7499 | 3.3032 | 1.0979 | 1.1771 | 1.1880 |
| | ACC↑ | **0.9913** | **0.9929** | 0.9953 | 0.9870 | 0.9947 | 0.9912 | 0.9736 | 0.9718 | **0.9833** |
| **Groot(WaveGrad)** | STOI↑ | 0.8261 | 0.8984 | 0.9759 | 0.9984 | 0.8587 | 0.9986 | 0.4585 | 0.4933 | 0.7534 |
| | MOSL↑ | 1.0449 | 1.1056 | 1.6318 | 4.5873 | 3.7324 | 3.1568 | 1.1034 | 1.1407 | 1.1665 |
| | ACC↑ | **0.9914** | **0.9951** | 0.9980 | 0.9806 | 0.9955 | 0.9905 | 0.9525 | 0.9795 | **0.9875** |
| **Groot(PriorGrad)** | STOI↑ | 0.8119 | 0.8867 | 0.9698 | 0.9997 | 0.8624 | 0.9999 | 0.4339 | 0.5238 | 0.7597 |
| | MOSL↑ | 1.0275 | 1.0591 | 1.4196 | 4.5581 | 3.5954 | 2.9838 | 1.0859 | 1.1358 | 1.1326 |
| | ACC↑ | **0.9952** | **0.9976** | 0.9977 | 0.8970 | 0.9373 | 0.9367 | 0.9847 | 0.9806 | **0.9868** |

bps, the evaluation metrics show only a marginal decrease. We also expanded the functionality of Groot to encompass Aishell-3 Chinese datasets [34], undertaking experiments to assess its cross-lingual fidelity and capacity. Detailed specifics are provided in the Appendix.

*The fidelity experiments conducted with different DMs on LJSpeech datasets are presented in Table 2.* These results reveal only a minor degradation in audio quality when employing WaveGrad across four different capacities, with the average STOI and SSIM scores remaining approximately 0.89 and 0.73, respectively. Moreover, recovery accuracy remains consistently high at 99%. In the case of PriorGrad, the average STOI and SSIM scores are 0.95 and 0.84, respectively. It sustains an exceptional accuracy rate across all capacities, averaging at 99.80%.

**Capacity** refers to the number of watermark bits that can be embedded. To evaluate the scalability of Groot in terms of watermark capacity, experiments were performed over a range of bps: 2500, 3000, 3500, 4000, 4500, and 5000 bps. Fig. 5 visualizes the impact on extracting accuracy as the watermark capacity increases. As observed, while LibriTTS exhibits minimal accuracy decline even at 5000 bps, the other two datasets experience noticeable decreases, albeit within an acceptable margin. Notably, from 2500 to 4500 bps, the proposed approach achieves an accuracy of approximately 99% with negligible impact on audio quality.

*5.3.2 The Comparison With SOTA Methods on Fidelity and Capacity.* Our evaluation commenced with a fidelity comparison between the proposed Groot and existing SOTA methods. The results are detailed in Table 3, with the capacity settings (16, 32, and 100 bps) conforming to the capacities reported in these SOTA methods. To effectively highlight the comparative outcomes, we introduce the prefix "D-" to denote the difference value between the watermarked audio and the original audio without watermark.

The analysis of the experimental results indicates that despite AudioSeal showing marginally smaller discrepancies in evaluation metrics, Groot consistently matches its audio quality. While Wav-Mark and Groot are nearly indistinguishable in terms of STOI, Groot presents smaller variations across other metrics. In comparison with DeAR and TimbreWM, Groot showcases superior audio quality. To further affirm the superior capacity performance of Groot, we compared with DeAR and TimbreWM at 2500 bps. The results distinctly show that the watermark extraction accuracy for both DeAR and TimbreWm drastically falls to 50%, highlighting their limitation in adapting to high-capacity conditions. In stark contrast, Groot, achieves high recovery accuracies of 99.04%, 99.70%, and 99.81% across different diffusion models, all the while preserving considerably good audio quality.

## 5.4 Robustness

*5.4.1 The Robustness of Proposed Groot and The Comparison With SOTA Methods Under Individual Attacks.* To ensure the integrity and resilience of our watermark against potential attacks during dissemination, we undertook comprehensive robustness experiments. It is essential to emphasize that our DMs, alongside our watermark encoder and decoder, are proprietary technologies and remain confidential. Consequently, attacks are constrained to execute under a black-box assumption, focusing primarily on post-processing manipulations of the content. In light of this, we assessed the robustness of the watermarked audio against various disturbances, including random Gaussian noise, low-pass and band-pass filtering, stretch, cropping, and echo. These specific attacks are elaborately detailed in the Appendix. Table 4 summarizes the results of robustness for Groot and four SOTA methods against individual attacks.

The experimental results validate the excellent robustness of the proposed Groot against individual attacks. Groot demonstrates superior robustness against Gaussian noise and echo compared to

**Table 5: Comparison of Robustness Under Compound Attacks.**

| Methods | Lowpass+Noise | | | Bandpass+Echo | | | Cropping+Stretch | | |
|---|---|---|---|---|---|---|---|---|---|
| | STOI↑ | MOSL↑ | ACC↑ | STOI↑ | MOSL↑ | ACC↑ | STOI↑ | MOSL↑ | ACC↑ |
| WavMark [1] | 0.9997 | 4.4625 | 0.5290 | 0.5653 | 1.1178 | 0.8315 | 0.4179 | 1.4192 | 0.8863 |
| DeAR [24] | 0.7562 | 4.2551 | 0.5067 | 0.7509 | 4.2743 | 0.9735 | 0.7617 | 4.3198 | 0.6950 |
| AudioSeal [33] | 0.8992 | 1.0808 | 0.5969 | 0.6570 | 1.1926 | 0.7135 | 0.4149 | 1.0878 | 0.6951 |
| TimbreWM [25] | 0.9022 | 1.1694 | 0.6058 | 0.6559 | 1.1876 | 0.8989 | 0.4135 | 1.5843 | 0.8983 |
| **Groot(DiffWave)** | 0.8540 | 1.0649 | **0.9803** | 0.7852 | 1.1991 | **0.9810** | 0.9366 | 1.0957 | **0.9615** |
| **Groot(WaveGrad)** | 0.8995 | 1.0929 | **0.9765** | 0.6552 | 1.1749 | **0.9821** | 0.4581 | 1.0964 | **0.9249** |
| **Groot(PriorGrad)** | 0.8858 | 1.0579 | **0.8434** | 0.6626 | 1.1410 | 0.9203 | 0.5393 | 1.0965 | **0.9166** |

| Methods | Noise+Echo | | | Noise+Bandpass | | | Noise+Bandpass+Echo | | |
|---|---|---|---|---|---|---|---|---|---|
| | STOI↑ | MOSL↑ | ACC↑ | STOI↑ | MOSL↑ | ACC↑ | STOI↑ | MOSL↑ | ACC↑ |
| WavMark [1] | 0.9997 | 4.4625 | 0.5300 | 0.9997 | 4.4625 | 0.5304 | 0.4967 | 1.0776 | 0.4907 |
| DeAR [24] | 0.7520 | 4.2693 | 0.9130 | 0.7503 | 4.2868 | 0.9257 | 0.7491 | 4.2860 | 0.9131 |
| AudioSeal [33] | 0.6875 | 1.0530 | 0.5886 | 0.7911 | 1.1084 | 0.6060 | 0.5999 | 1.0585 | 0.5904 |
| TimbreWM [25] | 0.6878 | 1.1927 | 0.5934 | 0.7934 | 1.0829 | 0.6281 | 0.5990 | 1.0881 | 0.5931 |
| **Groot(DiffWave)** | 0.7554 | 1.0864 | **0.9922** | 0.7154 | 1.0927 | **0.9924** | 0.7196 | 1.0526 | **0.9823** |
| **Groot(WaveGrad)** | 0.6777 | 1.0581 | **0.9826** | 0.7771 | 1.1136 | **0.9932** | 0.5892 | 1.0537 | **0.9760** |
| **Groot(PriorGrad)** | 0.6772 | 1.0358 | **0.9837** | 0.7719 | 1.0617 | **0.9332** | 0.5918 | 1.0387 | **0.9151** |

SOTA methods, particularly at noise levels of 5 dB and 10 dB. Here, it achieves watermark extraction accuracy exceeding 99% with negligible degradation. Even undergoing echo effect, its recovery precision maintains an impressive rate above 98%. Although DeAR method attains perfect accuracy against band-pass filtering and stretching, its accuracy markedly declines after cropping and at a noise level of 5dB, barely surpassing 70%. Despite Groot does not always secure the highest recovery accuracy for certain attacks, it maintains high average accuracies across different DMs-98.68%, 98.56%, and 96.82%, respectively. This solid performance of Groot further confirms its strong robustness against individual attacks.

*5.4.2 The Comparison With SOTA Methods Under Compound Attacks.* With unpredictable transmission environments and the possibility of malicious interventions during dissemination, content is likely to encounter complex, combined attacks in real-world scenarios. To delve deeper into the robustness of Groot with increased rigor, we evaluated its performance under five composite attacks, each integrating two distinct types of individual attacks, alongside one composite attack that combines three single attacks. The specific attack combinations are detailed as follows. 1) low-pass filtering succeeded by Gaussian noise, 2) band-pass filtering followed by an echo attack, 3) cropping and subsequently stretching, 4) Gaussian noise followed by echo, 5) Gaussian noise coupled with band-pass filtering and, 6) Gaussian noise succeeded by band-pass filtering coupled with echo.

The results of the robustness experiments against compound attacks are presented in Table 5. Concretely, Groot illustrates exceptionally superior robustness against compound attacks 1 and 4, with watermark extraction accuracy far surpassing other SOTA methods. Despite facing multifaceted challenges of compound attacks 4 and 5, the recovery accuracy maintains at 99.22% and 99.24%, respectively. Even when confronted with compound attack 6, which amalgamates three separate attacks, Groot sustains an accuracy level of over 90%, peaking at 98.23%. Although DeAR exhibits reasonable robustness against compound attack 2, it fails to maintain the anticipated robustness against compound attacks 1 and 3. In contrast, Groot consistently delivers commendable accuracy across

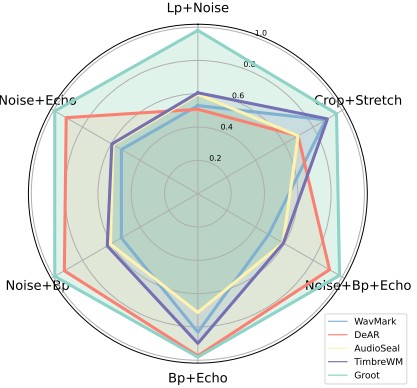

**Figure 7: Visualization for Robustness Against Compound Attacks of Groot Compared to SOTA Methods.**

all composite attacks, underscoring its well-balanced and dependable robustness. Furthermore, Fig. 7 utilizes a radar chart to visually represent the robustness of various SOTA methods in countering compound attacks. A more extensive area within the radar chart signifies enhanced robustness. The radar chart for Groot distinctly reveals a larger coverage area, highlighting its superiority over SOTA methods.

# 6 CONCLUSION

In this paper, we propose Groot, a novel generative audio watermarking method, aimed at effectively addressing the challenge of proactively regulating the generated audio via DMs. Groot instills the watermark into DMs, enabling the watermarked audio can be generated from the watermark via DMs. Leveraging our designed watermark encoder and decoder, there is no requirement for retraining the DMs. Robustness has been enhanced precisely due to the meticulous watermark decoder and jointly optimized strategy. The experimental results and comparisons to SOTA methods further illustrate that Groot exhibits potent and well-balanced robustness capable of countering individual and even compound attacks while ensuring superior fidelity and capacity. Regarding future work, we will introduce the noise layer and incorporate the more suitable watermark encoder and decoder to boost the fidelity and robustness of the generative watermarking method.

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
