# OpenReview forum: "GROOT: Generating Robust Watermark for Diffusion-Model-Based Audio Synthesis"
_acmmm.org/ACMMM/2024/Conference — MM2024 Poster_

### Official Review · Reviewer_VMNu · 2024-05-17

**Rating:** 2
**Confidence:** 4

**Summary:**

In this paper, a watermarking method for vocoder based on diffusion model is proposed. Specifically, the watermark is embedded by adding the watermark feature to the random initial latent code of diffusion vocoder before the waveform generation process, and the watermark is extracted from the generated waveform by the corresponding decoder. The identity of the model is verified by the extracted watermark. In this example, the process of watermark embedding and wavform generation occurs at the same time, and in the experiment, the proposed method outperforms the leading state-of-the-art post-processing based watermarking method.

**Strengths:**

- Innovatively introduce generative watermarking into the field of audio watermarking, and add watermarking to the audio vocoder. In recent years, researchers have conducted extensive research on model watermarking, but most of them used image generation model as an example to demonstrate the performance of the method, without exploring it in the audio scene. This paper attempts to explore generation model watermarking the audio field.
- The logic is clear and the paper is written in an easy-to-understand manner.
- The paper conducted a large number of experiments to explore the effectiveness of the method.

**Limitations:**

- Insufficient motivation description. Why is vocoder watermarking needed? It needs to be explained clearly, if the post-processing based audio watermarking is enough.
- Insufficient contribution. The proposed method seems to be a special case of diffusion model watermarking (like Tree-Ring[1]), just an application in the audio vocoder scenario.
- Appropriate methods are not cited (Tree-Ring and other general diffusion watermarking). As a general diffusion watermark, the methods can be directly used in diffusion-based vocoders (a special case) as a baseline.
- How is the Cropping test implemented? Through supplementary, it is found that the length of the audio will change due to cropping. According to the description of the network structure of the proposed method, the input and output sizes of the decoder should be fixed. So after cropping, how to extract the watermark?
- Lack of generality testing, no cross-testing of the watermark extractor between different vocoders, ie. traing on A and test on B, "plug-and-play" does not make sense.
- The regeneration process will cause great losses, the regenerated audio maybe very different from the original audio, which is bad for users. Why the "D-" indicator is used to measure audio quality. This seems unfair to the compared post-processing based watermarking methods. Moreover, the SSIM index of audio containing watermark will be strangely higher than that of audio without watermark, which shows that it is unreasonable to use "D-" (difference values) as an index.
- How does the proposed watermarking algorithm solve the problem of desynchronization? WavMark obtains desynchronization robustness through shift distortion in the noise layer to face distortions such as cropping and I found that TimbreWM mainly embedding watermark information in frequency to face this problem. I would like to revise the rating upon addressing this question.

[1] Tree-Ring Watermarks: Fingerprints for Diffusion Images that are Invisible and Robust

**Suitability:**

2

---

### Official Review · Reviewer_m3zj · 2024-05-20

**Rating:** 5
**Confidence:** 4

**Summary:**

A new method was proposed for embedding watermark information into the generated audio.

**Strengths:**

The embedding stage is merged with the generation stage. According to the special designed neural network structure, the comprehensive performance of the proposed watermarking method is improved. This method is based on the typical codec structure, and has certain innovation on embedding the watermark into the generated audio. The experimental setup is reasonable and the experiments are sufficient. The manuscript was well-written and organized, which is easy to follow for the readers.

**Limitations:**

1. It is necessary to clarify the respective roles played by the watermark embedding mode (generative watermarking), the encoder and decoder in the overall performance improvement, and better reflect the advantages of embedding watermark in the generation stage.
2. The further explanation is needed to clarify why the attack layer was not added during the training, but still achieved good results in the attack experiments.
3. In the last paragraph of Sec. 4.3, whether the subscript "Audio" of the loss function value refers to "Aud".
4. It seems that there are no experimental results of proposed Groot based on DiffWave in Table 2.
5. As shown in Figure 2, the noised latent is inputted into LDM during training, and the latent σ is inputted into LDM during inference. Why is the input for the inference process not noised latent? Can the distributions of the noised latent and the latent σ remain consistent?
6. The results of comparison experiment can only demonstrate the performance of the proposed method, the authors are suggested to show the relationship between the performance improvement and watermark embedding before audio generation. In other words, how does the watermark embedding before audio generation work on the performance improvement should be clearly justified and demonstrated.

**Suitability:**

2

---

### Official Review · Reviewer_7zau · 2024-05-21

**Rating:** 5
**Confidence:** 3

**Summary:**

This paper proposes a robust generative audio watermarking method coined Groot, which synchronizes the process of watermark generation and audio synthesis. Groot incorporates a dedicated watermark encoder, a parameter-fixed diffusion model, and a lightweight encoder. Experimental results validate that Groot exhibits remarkable performance in terms of watermark robustness, capacity, and audio quality.

**Strengths:**

The implementation of Groot can be a plug-and-play solution for any existing diffusion model. The training overhead is acceptable since it does not require retraining the diffusion model. It is a promising generative audio watermarking method tailored for diffusion models. The evaluation is also thorough, including robustness tests on various individual and compound attacks.

**Limitations:**

Although the proposed generative watermarking framework has no noise layer, it is surprisingly robust and can resist various common distortion attacks. However, I am concerned about whether the framework can withstand desynchronization attacks, such as pitch shift, time stretch, and jitter.

Some experimental results necessitate more detailed explanations. For example, the presence of negative values in Table 3, and the trade-off between audio quality and extraction accuracy at high capacities, as shown in Figure 5.

In the inference stage, it is unclear whether the embedded watermark bits can be changed flexibly. Moreover, I would like to know if the proposed watermarking paradigm can adapt to generative image watermarking schemes, even though they involve different media modalities.

**Suitability:**

2

---

### Official Review · Reviewer_cjMk · 2024-06-01

**Rating:** 5
**Confidence:** 2

**Summary:**

In this paper, the authors proposed a generative robust audio watermarking method to proactively supervise the synthesized audio and its source diffusion models. The idea is to design the lightweight encoder/decoder for watermarking. The idea is novel and the manuscript is well written.

**Strengths:**

The proposed method achieves a remarkable accuracy gain over the SOTA methods. Besides, the algorithm is described concisely and clearly with a good reproducibility.

**Limitations:**

The sigificance of using watermark for the generative audio synthesis should be further investigated. Is it used for copyright protection of the artificial content? It seems that the generative models with/without watermarking modul may yield the similar content. How to distinguish the ownership of the generative content derived from the other legal model and the proposed method?

**Suitability:**

3

---

### Meta-Review · Area_Chair_Di2v · 2024-07-02

**Recommendation:** Accept (Poster)
**Confidence:** 4

**Metareview:**

This paper presents Groot, which can serve as a plug-in to audio diffusion models to add robust watermarks. The reviewers find the approach achieves outstanding performance over prior work. There are some weaknesses that should be stated clearly. For example, the watermark bits are fixed during inference. Also, clarify how cropping is performed. The AC recommends acceptance of the paper and suggests the authors to revise the paper accordingly.